# Biochemical Methane Potential of Cork Boiling Wastewater at Different Inoculum to Substrate Ratios

Roberta Mota-Panizio [1,2], Manuel Jesús Hermoso-Orzáez [3,*], Luis Carmo-Calado [1], Gonçalo Lourinho [1] and Paulo Sérgio Duque de Brito [1]

1 VALORIZA—Research Center for Endogenous Resource Valorization, Polytechnic Institute of Portalegre, 7300-555 Portalegre, Portugal; rpanizio@ipportalegre.pt (R.M.-P.); luis.calado@ipportalegre.pt (L.C.-C.); goncalo.lourinho@ipportalegre.pt (G.L.); pbrito@ipportalegre.pt (P.S.D.d.B.)

2 MEtRICs-Mechanical Engineering and Resource Sustainability Center, Department of Science and Technology of Biomass, Faculty of Science and Technology, Universidade NOVA de Lisbon, 2825-149 Costa da Caparica, Portugal

3 Department of Graphic Engineering Design and Projects, Universidad de Jaen, 23071 Jaen, Spain

* Correspondence: mhorzaez@ujaen.es; Tel.: +34-610-389-020

**Abstract:** The present study evaluates the digestion of cork boiling wastewater (CBW) through a biochemical methane potential (BMP) test. BMP assays were carried out with a working volume of 600 mL at a constant mesophilic temperature (35 °C). The experiment bottles contained CBW and inoculum (digested sludge from a wastewater treatment plant (WWTP)), with a ratio of inoculum/substrate (Ino/CBW) of 1:1 and 2:1 on the basis of volatile solids (VSs); the codigestion with food waste (FW) had a ratio of 2/0.7:0.3 (Ino/CBW:FW) and the codigestion with cow manure (CM) had a ratio of 2/0.5:0.5 (Ino/CBW:CM). Biogas and methane production was proportional to the inoculum substrate ratio (ISR) used. BMP tests have proved to be valuable for inferring the adequacy of anaerobic digestion to treat wastewater from the cork industry. The results indicate that the biomethane potential of CBWs for Ino/CBW ratios 1:1 and 2:1 is very low compared to other organic substrates. For the codigestion tests, the test with the Ino/CBW:CM ratio of 2/0.7:0.3 showed better biomethane yields, being in the expected values. This demonstrated that it is possible to perform the anaerobic digestion (AD) of CBW using a cosubstrate to increase biogas production and biomethane and to improve the quality of the final digestate.

**Keywords:** cork boiling wastewater; anaerobic digestion; biogas; food waste; cow manure





## 1. Introduction

The cork oak (*Quercus suber* L.) is a native forest species protected by Portuguese legislation; it is distributed throughout the western part of the Mediterranean region. Historically, this primitive species has had greater importance in the Portuguese forest due to its wide distribution throughout the country and in the south. In terms of the manufacturing industry, Portugal's position is even more relevant since 60% of the raw material collected is processed by national industries, which export 90% of the processed cork [1–3]. Cork is mainly composed of suberin (between 40% and 50%), lignin (between 20% and 30%), polysaccharides (between 10% and 20%), and extracts (ceroids and tannins are between 10% and 15%). However, its composition depends on many variables such as climatic conditions, soil conditions, origin, dimensions, and age of trees [3,4]. The extracts present in cork are nonstructural organic components of the cellular wall. They include families of compounds such as n-alkanes, n-alkanols, waxes, fatty acids, triterpenes, glycerides, sterols, phenols, and polyphenols. Phenolic compounds represent about 90% of the contaminants present in CBW [5]. Among the acids present, ellagic acid was identified as the main phenolic compound [3]. The process consists of immersing the cork boards in hot water for up to 2 h to allow the material to acquire elasticity for manufacturing and also to clean and disinfect the boards [6].

It is estimated that in Portugal, around 34% of its total area is currently exploited, which corresponds to an average annual production of 150 thousand tons. The cork transformation process involves the use of large amounts of water and, consequently, high production of wastewater. It is estimated that 400 L of water is consumed per ton of cork processed in the raw material purification stage, known as cooking, where the raw material is purified by immersion in hot water (temperature 95 to 105 °C) for one hour. The 150 thousand tons of cork creates about $6.0 \times 10^7$ L of CBW. During the process of cooking cork boards, a wide range of organic compounds is transferred to the boiling water, thus resulting in a generation of wastewater with a high organic load and little biodegradability [7,8]. The compounds accumulated during the cooking process include phenolic acids, namely, gallic, vanillic, syringic, ferulic, and ellagic acids, in addition to tannins, 4,6-trichloroanisole, and pentachlorophenol. Carboxylic acids of polyphenols contribute to the acidification of the medium, which characterizes the effluent with a pH between 4 and 5 [9]. Although the cork boiling wastewater has a low biodegradability (low $BOD_5/COD$ ratio, between 0.3 and 0.45) due to the presence of polyphenols and tannins, currently, no specific treatments are applied to these waters before discharge into wastewater treatment plants (WWTPs) [10]. The environmental impact of this activity is worsened by the fact that the treatment of this effluent is difficult to achieve through conventional purification processes, namely, those based on biological treatment, due to the high concentration of organic charge, with a recalcitrant character normally present [8]. This problem is potentially even more serious because there are still many small industrial factories with low and discontinuous wastewater flows that are sent to the receiving WWTPs without any prior treatment [11]. In this sense, these effluents from the cork industry can constitute a serious environmental problem, namely, the contamination of soils and groundwater.

The most technologically advanced treatments for the effluent from the cork transformation process go through early removal of the organic matter through chemical coagulation and flocculation processes, followed by a biological process. In addition to these methods, membrane purification processes and chemical oxidation processes have also been used, with reagents such as Fenton or ozonation (advanced oxidation processes—POAs). Some of these processes can be combined to obtain better treatment results [12]. Below, we describe some studies dedicated to the treatment and recovery of cork effluents found in the literature.

Pontes-Robles et al. carried out combined tests of coagulation–flocculation/solar photo-Fenton/aerobic biotreatment to evaluate the microbiological action of cork cooking water. The results obtained demonstrated that there is an incompatibility in the use of biological treatment combined with the POA process [13]. Torres-Socías et al. studied the photo-Fenton process and solar treatment with ozone alone and in combination with hydrogen peroxide under different pH conditions on a pilot scale. The authors also evaluated the effect of physical–chemical pretreatments with different flocculants ($FeSO_4$ and $FeCl_3$). Although the physical–chemical pretreatment with Fe3+ provided good removal of COD and total suspended solids, it was found that the post-treatment with solar photo-Fenton did not improve. On the contrary, the ozone-based process was improved after the physical–chemical pretreatment with $Fe_3^+$, achieving greater degradation efficiency with less ozone consumption for the combination with $O_3$ at the initial pH 7.0. After treatment with solar photo-Fenton, toxicity and biodegradability remained constant at their initial values. The authors also performed the Zahn–Wellens test, which was used to study long-term biodegradability and a possible adaptation of biomass to the effluent after being partially treated with the photo-Fenton process [14]. The results showed decreases in toxicity values and increased short-term biodegradability for boiling wastewater treated with ozonation systems. Toxicity and biodegradability remained constant when compared to baseline values [10]. The use of membranes has also been explored as an alternative to physical–chemical treatments. The applications of this technology include microfiltration, ultrafiltration, and nanofiltration processes, being differentiated only by the pore size of

the membrane. Benitez et al. studied the use of ultrafiltration membranes for the treatment of wastewater from cork. With the study, they found that the effluent after treatment fell within the discharge parameters required by law [15].

In the literature, membrane technology is shown to be quite efficient; however, this treatment presents problems with the accumulation of sediments in the membrane, reducing the amount of effluent that is treated. Thus, the use of membrane technologies implies the performance of previous treatments, such as coagulation/flocculation, to remove suspended solids. Another limitation relates to the expenses associated with energy consumption and the monitoring of the process to ensure the smooth functioning of the system [16].

Ponce-Robles et al. performed several tests that demonstrated the characteristics of low biodegradability and the medium-to-low acute toxicity for the cork effluent, requiring biological treatments. Aerobic biological tests of wastewater from cork were carried out using mixed liquor from a municipal WWTP. The treatment evaluation parameters were based on optical microscopy, plaque counting, DNA extraction, and massive sequencing techniques. The results showed a reduction in the amount of total and volatile solids, concentration of DNA, general bacteria, and ammonia-oxidizing bacteria. It has been demonstrated that oxidative processes are necessary for partial remediation of the effluent, thus making it more biocompatible for the installation of a biological system so that the treatment of the effluent is at competitive operating costs [17].

Among biological treatments, anaerobic digestion (AD) is the most used due to the high rate of energy recovery and the low associated environmental impact [18]. Anaerobic digestion is a microbial process in which microorganisms break down biodegradable material in an oxygen-free environment to produce a solid digestate, together with biogas [19].

Anaerobic digestion is a promising technology for the treatment of effluents rich in organic compounds, also enabling the recovery of energy contained in the effluent through the production of methane. In this sense, Marques et al. studied the anaerobic digestion process to reduce the polluting organic load of cork effluent. The authors evaluated the methane production potential in batch effluent and mesophilic conditions (37 °C $\pm$ 1) for a period of 44 days. The acid effluent (pH 5.8) had an organic potential of 6.5 kg/m$^3$ of COD, rich in phenolic compounds (about 1 kg/m$^3$) and deprived of nitrogen (0.04 kg/m$^3$). With the tests carried out in concentrations of 3 kgCOD/m$^3$, the authors obtained a methane production of about 15 mgCOD-CH$_4$/m$^3$ after 15 days, and this was kept stable until the end of the experiments. For tests with a concentration of 6 kgCOD/m$^3$, the first 15 days of tests produced most of the methane (27 mgCOD-CH$_4$/m$^3$). They concluded that the cork effluent can be treated and valued by anaerobic digestion, with the direct application of biogas, while the digested flow can be used in cork oak forests as an organic supplement for the soil. The anaerobic digestion process can be considered part of an effluent treatment process for further application of technologies such as ultrafiltration [20].

The application of AD to industrial effluents with a low concentration of organic matter and rich in recalcitrant compounds, such as dioxins and the association of bacteria that can cause inhibition in the process, is under development. The improvement process to degrade certain toxic compounds is one of the main challenges for research. The introduction of cultures capable of degrading these effluents can be an efficient option for effluents that are difficult to degrade [21].

The selection of an appropriate substrate for the inoculum ratio (S/Ino) is critical to prevent volatile fatty acid (VFA) accumulation and accelerate the start of the process, with the intention of maximizing the methanogenic yield and the stability of the digestate. A very high S/Ino can overload microbial populations; in reverse, a lower value of S/Ino can result in high reactor volume requirements and less CH$_4$ (Dixon et al. 2018) [22]. Hobbs et al. [23] studied an S/Ino of 0.3 for food waste in the anaerobic digestion process and concluded that it was close to reaching CH$_4$ saturation at the end of the test but produced the least amount of CH$_4$ compared to ratios of 1:0 and 2:1; this aspect may be due to the ability to resist a change in pH (buffer capacity) of this effluent. Li et al. [24]

evaluated the combined effects of residual cooking oil content (33–53%) and food residues with different S/Ino ratios (0.5–1.2) on biogas yield, process stability, and organic parameter reduction during the anaerobic digestion process. The author concluded that S/Ino ratios greater than 0.70 caused inhibition and resulted in low biogas yields, and the digestate showed higher levels of propionic and valeric acids and high amounts of ammoniacal nitrogen from total and free ammonia nitrogen. Cremonez et al. (2019) [25] evaluated the level of decomposition of the material and volume of biogas produced in the AD process of cassava starch polymer (CSP) under different Ino/S ratios. Six different mixtures of Ino/S of 0.04, 0.08, 0.20, 0.60, and 1.00 and a control treatment without any addition were studied. The maximum values of SV removal found were greater than 85% in the treatment of 0.08 and greater than 90% in the treatment of 0.04. The treatment with a proportion of 0.08 resulted in the production of 1384 mL of biogas per gram of VSs removed. The author concluded that the maximum concentrations of methane in the treatments of 0.04, 0.08, 0.2, 0.6, and 1.0 were 87.09%, 84.48%, 81.45%, 74.97%, and 65, 83%, respectively. Another study investigated the influence of different Ino/S ratios (0.5:1 to 4:1) during anaerobic codigestion under mesophilic conditions of spent coffee grounds and cow manure. Methane yields gradually increased from the Ino/S ratio of 0.5:1 to 3:1, and the highest methane yield (225 mLCH4 gVS$^{-1}$) was achieved in the I:S ratio of 3:1. Lower methane percentage was obtained in the ratios of 3.5:1 and 4:1. Unstable process conditions were established in the lowest examined Ino/S ratio (0.5:1), which caused the accumulation of volatile fatty acids [26]. Córdoba et al. (2018) [27] studied the yield of methane production from swine wastewater, using sewage sludge as an inoculum in three S/Ino ratios, called A (1:1), B (3:1), and C (6:1). The highest biogas yield that the author obtained in treatment was with the A ratio (554 ± 75 mL/g volatile solid (VS)). Cumulative methane production decreased from 382 ± 22 to 232 ± 5 mL/g VS when S/Ino increased from 1:1 to 6:1. The best performance in terms of kinetic parameters was obtained for Treatment A; however, Treatment B could still guarantee a stable process. The use of a higher concentration of inoculum generated a 463.1% higher methane production rate and required a 77.3% shorter adaptation time (latency phase) in the studied S/Ino range. Pagés-Diaz et al. (2020) [28] also studied the importance of the ratio S/Ino in the anaerobic digestion of liquid fractions of mixed biomass treated cohydrothermally and the organic fraction of solid urban residues. The study showed that the S/Ino ratio had a significant impact on $CH_4$ production yield, with the 1:3 ratio reducing methane production and the 1:2 ratio achieving the best results.

Cork effluent is very problematic due to the acidic character and constituents of low degradability, thus making it difficult to treat. Currently, few studies of cork boiling wastewater treatment by AD have been carried out, and information is very limited. The authors are only aware of a previous study that evaluated the AD of this effluent in order to enhance the reduction of organic pollutants [20]. Based on the considerations above, the present work aims to investigate the effect of two different ratios of Ino/CBW (1:1 and 2:1), Ino/CBW:FW (2/0.7:0.3 and 2/0.5:0.5), and Ino/CBW:CM (2/0.7:0.3 and 2/0.5:0.5) on the yield of accumulated methane production and the degradation of VSs.

## 2. Materials and Methods

The CBW used for anaerobic digestion tests comes from an association of industries located in the San Vicente de Alcántara area, Spain. The city is located in the northwest of the province of Badajoz. The region is considered the city of cork, where there is the highest concentration of manufacturing industries for this material; part of its culture and Celebrations are related to this activity [29].

The effluents produced by the cork industries are collected by the San Vicente de Alcántara city council and sent for treatment. The effluent studied was collected directly from a truck that arrived at the treatment plant. The CBW was obtained from mixtures from different companies in the region.

The inoculum (Ino) used for the BMP tests was obtained from an anaerobic digester in a wastewater treatment plant in a city in central Portugal. The treatment unit has an anaerobic digester in operation, and the Ino was collected from the end of the AD process, thus containing large amounts of bacteria and facilitating the start of operation of the studied digesters. Ino is an established digest, ready to be used in other AD tests or to be treated at the wastewater treatment plant.

Food waste (FW) and cow manure (CM) were used as cosubstrates for CBW co-management tests. The FW used came from the process of preparing homemade meals. Vegetable scraps (broccoli, cauliflower, cabbage, and potatoes), fruits (banana peel and watermelon), and scraps of chicken meat were used. The CM used was collected at the Portalegre Farmers Association, Portugal, where the association organizes animal auctions every week.

### 2.1. Characterization of Materials

A Hanna HI 9810 Portable pH/EC/TDS meter was used to measure pH, total dissolved solids, and conductivity. Before measurements, calibrations were performed, and the Standard Methods 4500-H+B method was followed. The determination of total solids (TSs) was performed using Standard Methods section 2540B. For the determination of volatile solids (VSs), we used Standard Methods section 2540E.

The determination of the elemental amount of C, H, N, S, and O was carried out using a ThermoFisher Scientific Flash 2000 CHNS-O analyzer and expressed in percentage. The elementary analysis is based on a destructive technique, introduced into the reactor and having undergone combustion at 900°C, being destroyed during analysis. The final result is a chromatogram that reflects the concentrations present in the analyzed material.

To determine chemical oxygen demand (COD), the APHA 5220D method was used. The process consisted of performing digestion in an AL 32 digester block, with $H_2SO_4$ and $K_2CrO_7$ at 150 °C for 2 h. For the measurements, a Turner Model 690 spectrophotometer with a wavelength of 600 nm was used.

Thermogravimetric analysis or thermal gravimetric analysis (TGA) analysis was performed using a thermogravimetric analyzer (PerkinElmer STA6000), simultaneously recording thermogravimetric measurements and differential thermal analyses (DTAs). The thermogravimetric tests of the samples (between 3.5 and 4.5 mg) were performed at 30 to 995 °C at a heating rate of 30 °C.min$^{-1}$, in atmospheric air. Fourier transformed infrared (FTIR) spectra were obtained as an average of 64 scans collected at a resolution of 16 cm$^{-1}$ using an ATR-FTIR spectrometer (Thermo Scientific Nicolet iS10) in the range of 4000 to 400 cm$^{-1}$. The powder samples were placed on the ATR crystal and compacted using a vertical screw to the plane for analysis. The differences in the spectral peaks of the different residues were then evaluated.

### 2.2. Experimental Tests

BMP tests were performed in 1000 mL glass bottles (Schott-Duran) with a 600 mL working volume. The reactors were placed in a water bath at 38 ± 1 °C (mesophilic conditions) and kept at a constant temperature [30]. The average temperature inside each one was considered to be around 35 °C. At the beginning of the process, each reactor was fed with adequate amounts of CBW and inoculum, with Ino/S ratios of 1:1 and 2:1, Ino/CBW:FW ratios of 1/0.7:0.3 and 2/0.5:0.5, and Ino/CBW:CM ratios of 1/0.7:0.3 and 2/0.5:0.5 on a VS basis. Subsequently, the reactors were "washed" with an inert gas (argon) for periods of 1 min before being sealed with 5 mm thick silicone discs and closed with a plastic screw cap, both from Schott Duran (produced in Germany).

Biogas production was measured indirectly from the reactor mass loss, as detailed by Hafner et al. [31]. The mass of each reactor was determined before and after biogas production periods. The biogas was removed/collected in Tedlar bags by perforating the silicone discs with the aid of a hypodermic needle until atmospheric pressure/equilibrium was reached.

The biogas collected was immediately characterized by a portable gas analyzer (Gas-Data GFM406), which allows the measurement of the percentages of the volume of $CO_2$, $CH_4$, $O_2$, $H_2S$, and CO in the mixture. The flasks were shaken manually at regular times.

The mixing was performed manually during the incubation period at regular times. The BMP test was terminated when biogas production ceased or had biogas production below 0.5% of accumulated production. The values shown were expressed using standard temperature and pressure.

## 3. Results and Discussion

### 3.1. Characterization of Substrate

Table 1 describes the characterizations performed on the substrate and inoculum used in the AD process. CBW has an acidic character, with a pH of 4.6. The Ino used was neutral, helping the CBW to reach the optimum pH (between 6.5 and 7.5) so that the AD process occurred favorably. CBW is rich in phenolic–polyphenolic compounds, which characterizes them as an acid effluent, with an acid pH between 4.5 and 5.5 [32–34].

**Table 1.** Characterization of the substrate, inoculum, and cosubstrates used.

| Parameter | Unit. | CBW | Ino | FW | CM |
|---|---|---|---|---|---|
| pH | - | 4.6 ± 0.2 | 6.7 ± 0.2 | 6 ± 0.2 | >> |
| Dissolved solids | mg/L | 740.00 ± 60 | 2840.00 ± 120 | 4440.00 ± 180 | >> |
| Conductivity | μS/cm | 1480.00 ± 350 | 5770.00 ± 980 | 8910.00 ± 1080 | >> |
| C | (%) | 44.17 ± 0.43 | 35.08 ± 2.08 | 46.29 ± 1.67 | 39.56 ± 0.94 |
| H | (%) | 6.03 ± 0.16 | 6.06 ± 0.33 | 6.43 ± 0.37 | 5.91 ± 0.31 |
| N | (%) | 3.7 ± 0.57 | 7.52 ± 0.52 | 4.79 ± 1.67 | 3.44 ± 0.16 |
| S | (%) | 0 | 0 | 0 | 0 |
| O | (%) | 29.96 ± 0.64 | 11.42 ± 0.38 | 28.74 ± 0.55 | 27.31 ± 0.35 |
| Ratio C/N | - | 12.06 ± 1.74 | 4.66 ± 0.05 | 9.66 ± 0.59 | 11.51 ± 0.05 |
| COD | mg/L | 7060.00 ± 100 | 1639.50 ± 13.75 | 82,975.00 ± 1500 | 84,475.00 ± 1062 |
| Total solids | mg/L | 8375.00 ± 25 | 14,525.00 ± 575 | 79,575.00 ± 3525 | 187,250.00 ± 1150 |
| Total solids | (%) | 0.87 ± 0.00 | 2.00 ± 0.72 | 6.9 ± 0.02 | 27.96 ± 0.86 |
| Volatile solids | mg/L TS | 6875.00 ± 25 | 9500.00 ± 300 | 68,925.00 ± 3225 | 136,600.00 ± 2400 |
| Volatile solids | (% TS) | 82.09 ± 0.05 | 65.43 ± 0.52 | 86.60 ± 0.22 | 729.00 ± 0.83 |

The CBW substrate used had a COD of 7060.00 mg/L, which is in accordance with the literature. In some previous works, CBW effluent presented COD with very wide values, varying between 2600.00 and 12,600.00 mg/L [32,35]. These values are dependent on the quality of the boards, the number of times the water is reused, and the location of the cork oak forest, among other factors that can change the characteristics of the raw material.

According to Pintor et al. [36] and Gonçalves et al. [37], the conductivity of CBW varies between 1200 and 1500 μS/cm. Additionally, according to Gonçalves et al. [37], the total solids of the effluents are in the order of 5200 mg/L.

In their tests, Ponce-Robles et al. used the effluents from the associated plant of the cork industries of San Vicente de Alcântara. Despite being from different lots, the pH (5.0) and conductivity (1684 μS/cm) values were relatively close. The authors characterized the amounts of polyphenols present in the effluent, being in the order of 455 mg/L. As they were effluents from the same region and produced by the same companies, it is possible that despite being from different lots, the conditions were as similar as possible compared to other effluents.

According to Sousa et al., bovine manure has a high COD and a high amount of total solids. Those values are very close, in the order of 84,000.00 mg/L of COD and 110,000.00 mg/L of total solids [38].

The characteristics of food waste vary widely according to the components that are included in the waste. These variations make it difficult to compare with works carried out by other authors.

The C/N ratio reflects the availability of nutrients in CBW, Ino, FW, and CM for the growth of bacteria. The low C/N ratio shows that the microorganism does not have adequate amounts of nutrients, especially carbon, for them to develop. Wang et al. (2014) reported that the low C/N ratio can result in the formation of large amounts of ammonia, which is toxic to the bacterial population since nitrite accumulates during the processes of biological denitrification, thus, causing the inhibition of the process [39]. The C/N ratio has a great influence on the competition between the dissimilar reduction of nitrate to gaseous products (denitrification) and ammonification. Therefore, when it comes to the availability of organic carbon, regardless of the form found in the digestate, it is important that the C/N ratio is observed.

In most animal waste, there is enough carbon for denitrification because there is a large amount of organic carbon available, that is, a high C/N ratio. Codigestion has several advantages, mainly related to the optimization of the C/N ratio, increased biodegradable organic load, dilution of potentially toxic compounds, and balance of available nutrients.

### 3.2. Production of Biogas and Biomethane

The accumulated production of biogas can be seen in Figure 1; the production started in the first days. The solids for both ratios were reduced, with the production of biogas derived from the solids added to the reactor.

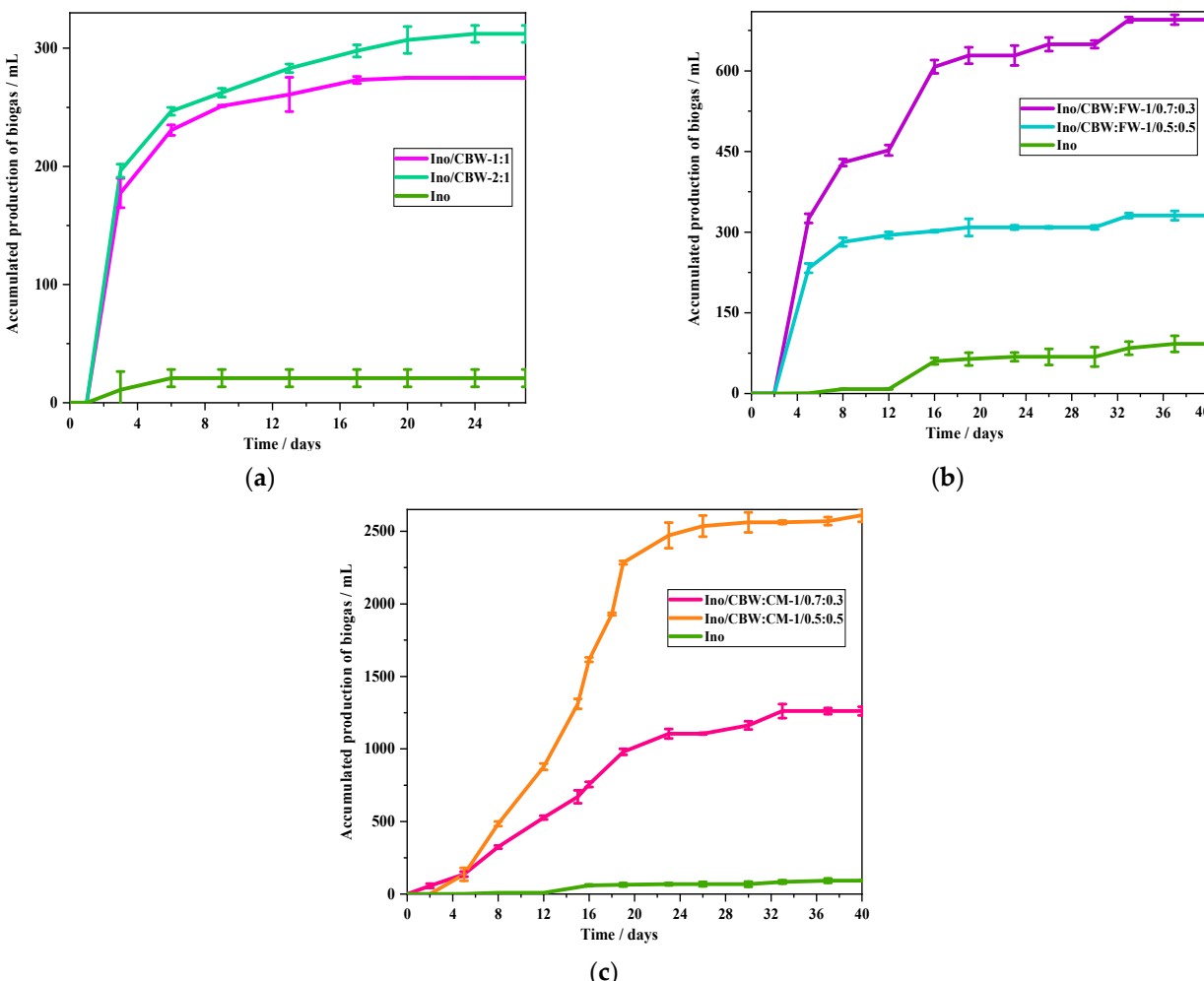

**Figure 1.** Accumulated biogas for the tests (**a**) anaerobic digestion (AD) of different ratios of mixture of inoculum and cork boiling wastewater (Ino/CBW); (**b**) anaerobic digestion of different ratios of mixture of inoculum, cork boiling wastewater, and food waste (Ino/CBW:FW); (**c**) anaerobic digestion of different ratios of mixture of inoculum, cork boiling wastewater, and cow manure (Ino/CBW:CM).

For each ratio studied, the tests were performed in duplicate. For the Ino/CBW 1:1 ratio, biogas production was $274.5 \pm 0.5$ mL, and the production in relation to the added volatile solids was $99.94 \pm 0.18$ mL.g.SV$_{added}$$^{-1}$. Regarding the 2:1 ratio, biogas production was $317.43 \pm 7.48$ mL, and the production in relation to the added volatile solids was $155.60 \pm 3.67$ mL.g.SV$_{added}$$^{-1}$. Compared with the literature, these values are much lower than expected.

Food waste has the characteristic of having good biodegradability. In order to have a good performance in relation to the degradation of organic matter, the system needs enough nutrients and minerals for the correct development of microorganisms to occur. Inhibitions can occur when digesting large amounts of food waste for long periods of operation. The reasons for the occurrence of inhibition are related to the imbalance of nutrients during the fermentation process—insufficient elements (zinc, iron, molybdenum, among others) and excess micronutrients (sodium, potassium, among others) [40]. In order to avoid problems arising from inhibition, several authors have suggested that FW with two or more substrates be used in the process in order to achieve better performance with the mixtures. The substrates must compensate for the nutritional deficiencies that the FW has, in addition to helping the growth of microorganisms [41–43].

For the tests carried out, the accumulated biogas for the tests with both mixtures (2/0.7:0.3 and 2/0.5:0.5) can be seen in Figure 1. For the ratio of 2/0.7:0.3 of Ino/CBW:FW, the production of biogas was 695.12 mL, and specific production in relation to VSs was 101.04 mL g.VS$_{add}$$^{-1}$. For the ratio of 2/0.5:0.5 of Ino/CBW:FW, accumulated biogas production was 330.83 mL, and production in relation to VSs was 32.98 mL g.VS$_{add}$$^{-1}$. It is possible to note that the biogas production for the second test was lower, indicating that an inhibition may have occurred, as previously mentioned.

Cow manure is used as fertilizer in the fields; however, the deposition of this material without pretreatment increases the potential for contamination and puts the health of the animals at risk [44]. The use of treatments that aim to reduce contaminants is of fundamental importance for the production of fertilizers from this type of waste [45]. The use of CM in anaerobic digesters is an alternative to potentiate an effluent with low methane production potential. Codigestion can increase biogas production by approximately 44% [46].

Figure 1 shows the accumulated volumes of biogas for the BMP tests of Ino/CBW:CM codigestion in the ratios of 2/0.7:0.3 and 2/0.5:0.5, with a duration of 40 days. For the 2/0.7:0.3 ratio tests, accumulated biogas production was 1261.15 mL, and cumulative production in relation to VSs was 372.76 mL.g.SV$_{added}$$^{-1}$. For the ratio of 2/0.5:0.5, accumulated biogas production was greater than 107% in relation to the test with a ratio of 2/0.7:0.3. The accumulated biogas was 2611.23 mL, with an accumulation in relation to VSs of 533 mL.g.SV$_{added}$$^{-1}$.

The volume of methane (Figure 2) production for the 1:1 ratio is $161.95 \pm 11.60$ mL and for the 2:1 ratio, $202.35 \pm 4.15$ mL; these values can be observed in Figure 2. The respective methane yields, calculated by dividing the final methane production volumes by the weight of the VSs added to the reactors, are $58.89 \pm 4.22$ and $99.18 \pm 2.03$ mL.g.VS$_{added}$$^{-1}$.

Figure 4 shows methane accumulation during the 40 days of the BMP test. When compared to the tests carried out in the previous chapter, it is possible to verify that FW is not viable as a cosubstrate for CBW AD.

The accumulated volume of methane production for the Ino/CBW:FW test with a ratio of 2/0.7:0.3 was 192.73 mL and 28.02 mL.g.SV$_{added}$$^{-1}$. For the ratio of 2/0.5:0.5, accumulated methane was 27.61 mL and 2.71 mL.g.SV$_{added}$$^{-1}$. These values demonstrate the inhibition of the process and the nonviability of comanagement with FW.

The BMP tests demonstrated that the codigestion of CBW with CM increased the potential for biogas production and the volume of methane. For the ratio of 2/0.7:0.3, accumulated methane production was 545.20 mL, and accumulated production in relation to the amount of VSs was 143.84 mL.g.SV$_{added}$$^{-1}$. For the 2/0.5:0.5 ratio, methane accumulation was 1002.91 mL, and accumulated production in relation to the amount of VSs was 205.15 mL.g.SV$_{added}$$^{-1}$. The accumulated methane values can be seen in Figure 2.

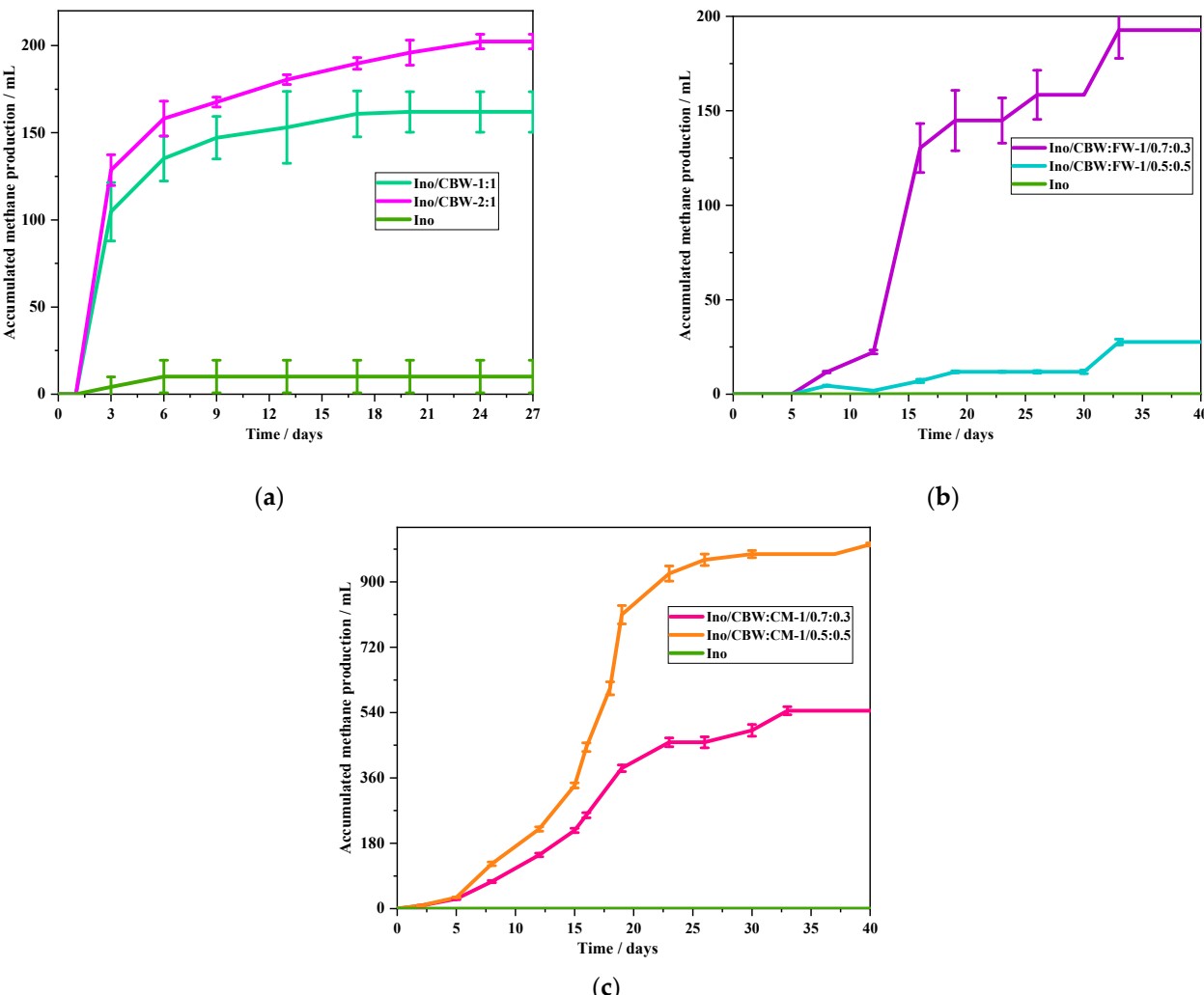

**Figure 2.** Accumulated of methane for CBW digestion tests (**a**) AD of different ratios of mixture of inoculum and cork boiling wastewater (Ino/CBW); (**b**) anaerobic digestion of different ratios of mixture of inoculum, cork boiling wastewater, and food waste (Ino/CBW:FW); (**c**) anaerobic digestion of different ratios of mixture of inoculum, cork boiling wastewater and cow manure (Ino/CBW:CM).

### 3.3. Characterization of the Digestate

The characterizations of the digestate of each vial are shown in Table 2. The pH of the digested ones was around 6.7, both for the different ratios and for the digester with the inoculum that was used for control. The dissolved solids for the studied mixtures were between 1700 and 2100 mg $L^{-1}$. The final C/N ratio in the reactors was between 7.7 and 8.7. The amount of final solids for the 1:1 ratio was 9850.00 mg $L^{-1}$; for the 2:1 ratio, the amount of solids was 11,625.00 mg $L^{-1}$.

The BMP tests for CBW codigestion with FW and CM, as well as the characterizations of the digestate, are shown in Table 1. For the codigestion tests, the previous tests were used; the Ino/CBW ratio of 2:1 was used in the assay. With the determination of the initial ratio, the substrate part was then tested in two different ratios of 2/0.7:0.3 and 2/0.5:0.5 for Ino/CBW:FW and Ino/CBW:CM.

The assays using CBW and FW, the final digested ones, maintained a pH between 7.3 and 6.0 for the ratios of 2/0.7:0.3 and 2/0.5:0.5, respectively. The final effluents had a reduction of 60.82% and 56.76%, relative to the total solids present in the digested mixtures.

Table 2. Characterization of the digestate of the different ratios.

| Parameter | Unit. | Ino/CBW-1:1 | Ino/CBW-2:1 | Ino/CBW:FW-1/0.7:0.3 | Ino/CBW:FW-1/0.5:0.5 | Ino/CBW:CM-1/0.7:0.3 | Ino/CBW:CM-1/0.5:0.5 | Ino Control |
|---|---|---|---|---|---|---|---|---|
| pH | - | $6.7 \pm 0.2$ | $6.7 \pm 0.2$ | $7.3 \pm 0.2$ | $6 \pm 0.2$ | $7 \pm 0.2$ | $7.1 \pm 0.2$ | $6.65 \pm 0.25$ |
| Dissolved solids | mg/L | $1720.00 \pm 200$ | $2065.00 \pm 150$ | $4440.00 \pm 200$ | $4650.00 \pm 250$ | >> | >> | $715.00 \pm 50$ |
| Conductivity | µS/cm | $3420.00 \pm 180$ | $4185.00 \pm 200$ | $8890.00 \pm 330$ | $9220.00 \pm 360$ | >> | >> | $1415.00 \pm 100$ |
| C | (%) | $37.63 \pm 1.38$ | $38.15 \pm 0.61$ | $35.24 \pm 0.8$ | $39.45 \pm 0.2$ | $35.71 \pm 0.45$ | $35.58 \pm 0.82$ | $35.31 \pm 1.13$ |
| H | (%) | $5.76 \pm 0.33$ | $6.33 \pm 0.13$ | $6.86 \pm 0.31$ | $7.36 \pm 0.48$ | $6.06 \pm 0.35$ | $6.39 \pm 0.66$ | $6.35 \pm 0.16$ |
| N | (%) | $4.34 \pm 0.15$ | $4.90 \pm 0.16$ | $6.53 \pm 0.12$ | $6.00 \pm 0.14$ | $4.77 \pm 0.09$ | $5.26 \pm 0.12$ | $5.02 \pm 0.06$ |
| S | (%) | 0 | 0 | 0 | 0 | 0 | 0 | 0 |
| O | (%) | $17.91 \pm 0.82$ | $12.27 \pm 0.91$ | $4.73 \pm 0.25$ | $14.88 \pm 0.54$ | $15.97 \pm 0.84$ | $15.00 \pm 0.28$ | $13.01 \pm 0.65$ |
| C/N ratio | - | $8.68 \pm 0.18$ | $7.79 \pm 0.17$ | $5.4 \pm 0.25$ | $5.68 \pm 0.19$ | $7.49 \pm 0.21$ | $7.34 \pm 0.18$ | $7.03 \pm 0.14$ |
| COD | mg/L | $7547.50 \pm 75$ | $7328.75 \pm 106$ | $2147.5 \pm 90$ | $8053.75 \pm 82$ | $19,982.50 \pm 365$ | $15,945.00 \pm 330$ | $1110.00 \pm 37.5$ |
| Total solids | mg/L | $9850.00 \pm 550$ | $11,625.00 \pm 775$ | $11,650.00 \pm 2600$ | $19,000.00 \pm 50$ | $27,200.00 \pm 200$ | $49,400.00 \pm 1000$ | $5900.00 \pm 2700$ |
| Total solids | (%) | $0.99 \pm 0.05$ | $1.18 \pm 0.07$ | $1.33 \pm 0.3$ | $2.17 \pm 0.12$ | $3.46 \pm 0.11$ | $5.51 \pm 0.03$ | $0.60 \pm 0.27$ |
| Volatile solids | mg/L VS | $8050.00 \pm 300$ | $8100.00 \pm 200$ | $5600.00 \pm 400$ | $10,450.00 \pm 250$ | $17,700.00 \pm 100$ | $33,600.00 \pm 2000$ | $2425.00 \pm 675$ |
| Volatile solids | (wt %) | $82.30 \pm 5.9$ | $70.00 \pm 2.99$ | $48.08 \pm 3.64$ | $55.02 \pm 1.89$ | $65.01 \pm 0.11$ | $68.00 \pm 2.67$ | $48.8 \pm 12.1$ |

From the BMP tests using CBW in codigestion with CM, it is possible to verify the stability relative to the pH for both proportions studied, with the pH at about 7.0. The total solids present in the final effluents had removals of 20.37% and 49.49% for 2/0.7:0.3 and 2/0.5:0.5, respectively.

### 3.4. Thermogravimetric and Spectroscopic Analysis

The TGA and DTG analyses make it possible to know for each sample the weight loss that occurs in a given temperature range during its combustion. The higher the temperature at which weight loss occurs, the more resistant and structurally organized is the organic fraction that is burning. Comparing different groups with different degrees of stability, according to the temperature changes in which the main weight loss occurs during combustion, should be indicative of the characteristics of the organic fraction. On the other hand, in comparison with the initial sample, whenever a weight loss disappears in the TGA and DTG profiles, a temperature at which it was presented in the crude analyses, this may indicate that the corresponding organic fraction that was being burned was mineralized due to the stabilization process. This weight loss may indicate the AD process is efficient [47–49].

The curves of mass loss as a function of temperature for the substrate used and the different cosubstrates are described in Figures 3 and 4; the differential thermal analyses are presented.

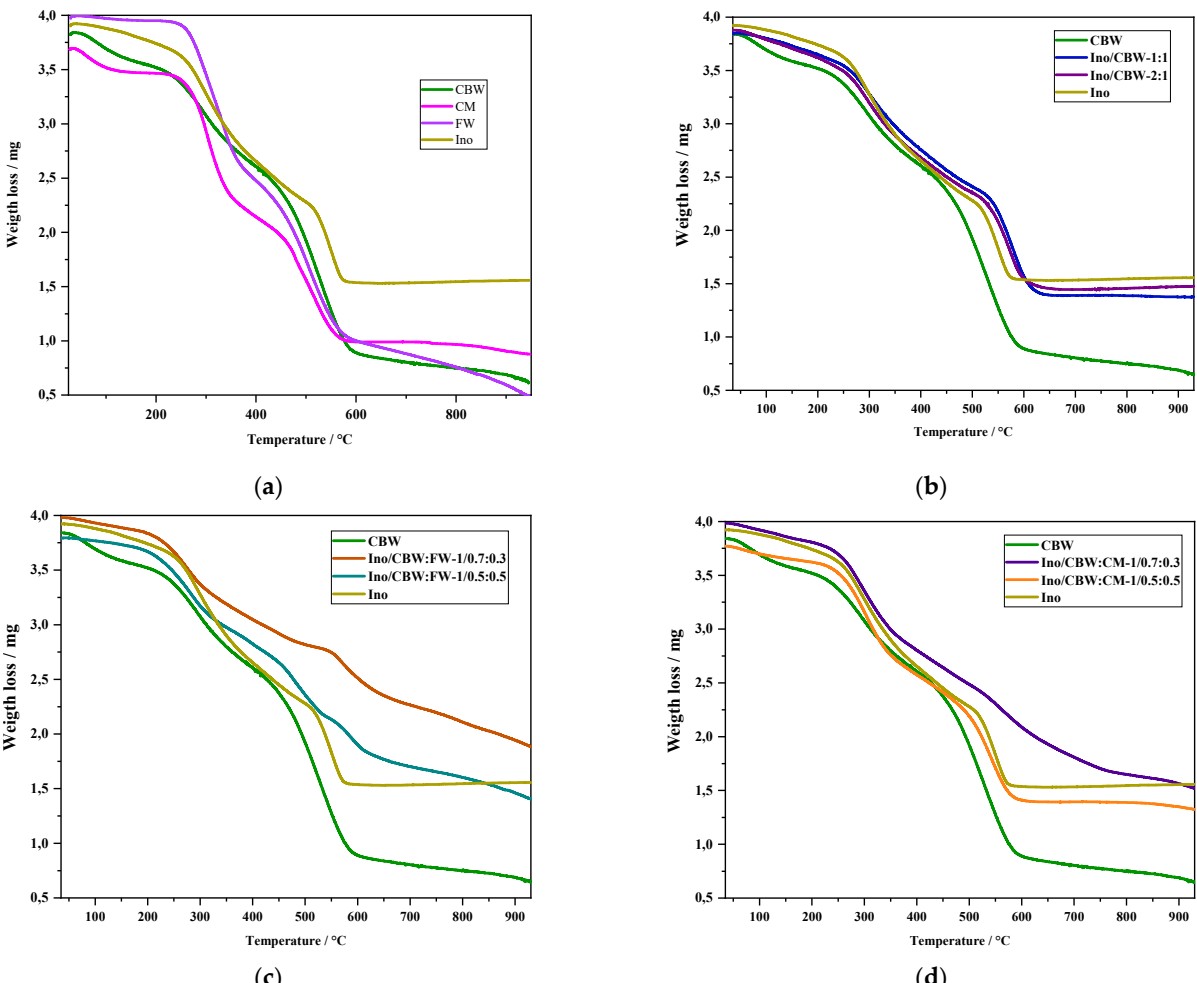

**Figure 3.** Thermogravimetric analysis. (**a**) Cork boiling wastewater (CBW), cow manure (CM), food waste (FW), and inoculum (Ino); (**b**) AD of different ratios of mixture of inoculum and cork boiling wastewater (Ino/CBW); (**c**) anaerobic digestion of different ratios of mixture of inoculum, cork boiling wastewater, and food waste (Ino/CBW:FW); (**d**) anaerobic digestion of different ratios of mixture of inoculum, cork boiling wastewater, and cow manure (Ino/CBW:CM).

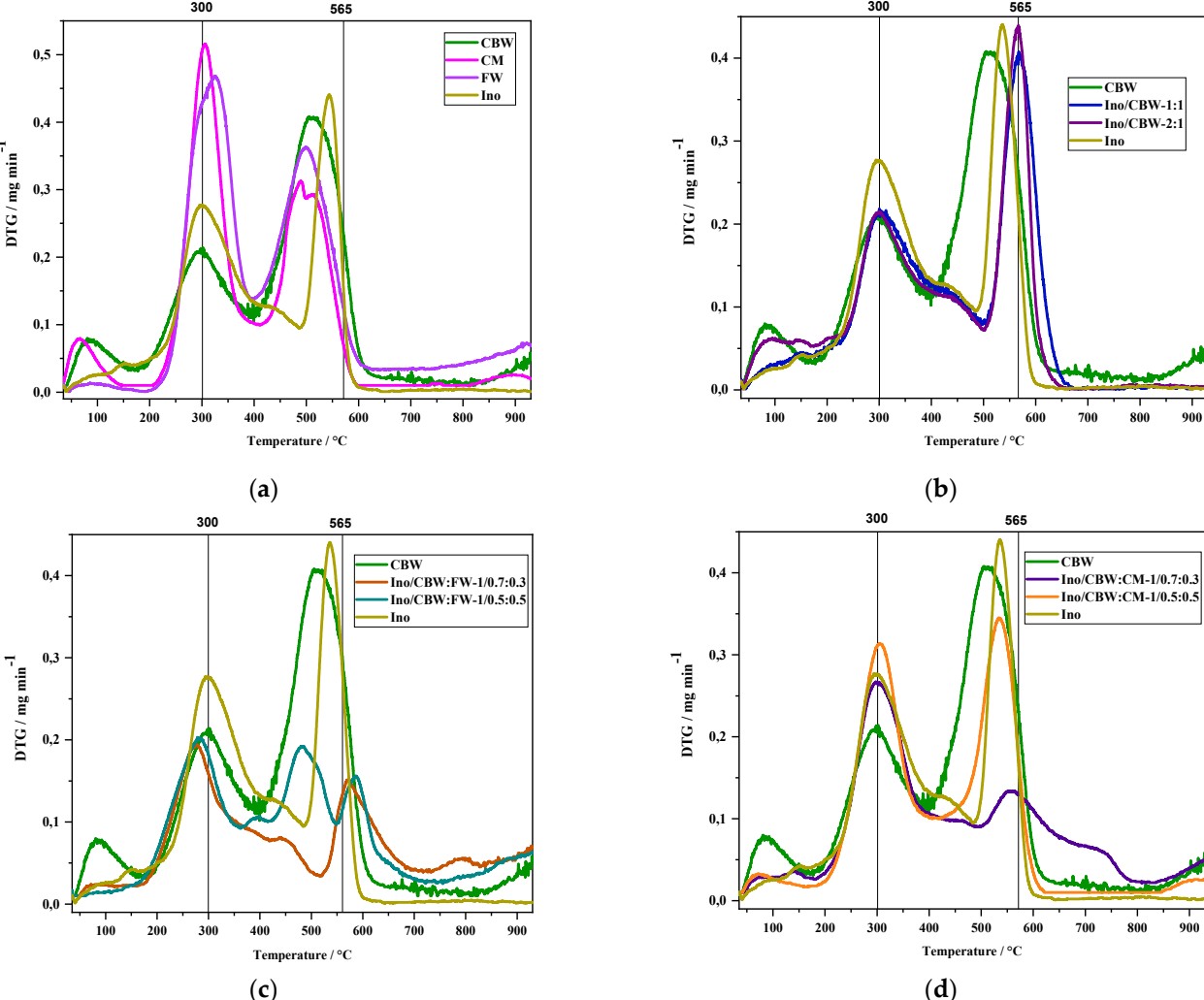

**Figure 4.** Differential thermal analysis. (**a**) Cork boiling wastewater (CBW), cow manure (CM), food waste (FW), and inoculum (Ino); (**b**) AD of different ratios of mixture of inoculum and cork boiling wastewater (Ino/CBW); (**c**) anaerobic digestion of different ratios of mixture of inoculum, cork boiling wastewater, and food waste (Ino/CBW:FW); (**d**) anaerobic digestion of different ratios of mixture of inoculum, cork boiling wastewater, and cow manure (Ino/CBW:CM).

Disregarding the first peak presented in the TGA and DTG, found between 50 and 150 °C (which represents the dehydration reactions [50]), the thermogravimetric profile can be divided into two main stages. The first big step (at about 280 °C) corresponds to easily degradable materials, carbohydrates, and semivolatile compounds. The second (up to approximately 450 °C) corresponds to organic polymers that were already present in the original material or are generated from the process. The loss of mass above 450 °C is related to complex organic material [51]. The weight loss recorded between 520 and 650 °C is related to the dissociation and decomposition of aromatic structures and high molecular weight polynuclear systems [52].

The first stage of mass loss in the TGA is practically imperceptible for the digestion tests with FW and CM. This first peak is exothermic and can be attributed to the thermal combustion of polysaccharides, decarboxylation of acid groups, and dehydration of aliphatic structures [53]. The transformation and decrease in the concentration of these substances during digestion cause a peak shift in the DTG towards lower temperatures and a decrease in intensity, which indicates that the availability of materials for the degradation of microorganisms is decreasing [49]. In the CBW AD process, no peak reduction corresponding to aromatic compounds was detected.

Two main zones are clearly identified in the DTG (Figure 4). The first peak associated with TGA is associated with the release of water, around 100 °C. According to Gómez et al., the peak in the DTG (close to 300 °C) corresponds to the materials of easy oxidation, and this peak is inversely proportional to the peak close to 550 °C. The higher temperature peak decreases as the process reaches stabilization [49]. The third peak in the sample, at 550 °C, corresponding to lignin, may be related to the presence of long-chain hydrocarbons and N compounds that contribute to thermal reactions at high temperatures [53], also representing the oxidation of aromatic structures and organic compounds with great thermal stability [52].

None of the DTG tests for the Ino/CBW ratio of 1:1 and the Ino/CBW ratio of 2:1 had reductions (if compared with the crude tests) in mass speed as the temperature increased. The 300 °C peak increases the degradation area of easily oxidized materials and reduces the peak area by 550 °C. The second peak decreases as the stabilization of the process occurs [49], being verified as soon as the process has not reached stabilization.

For the tests of Ino/CBW:FW (2/0.7:0.3 and 2/0.5:0.5) and Ino/CBW:CM (2/0.7:0.3), the peak around 480 to 580 °C was reduced and, consequently, the peak at about 280 to 310 °C increased. The Ino/CBW:CM ratio 2/0.5:0.5 test was the least stable at the end of the process. The rest remained very close.

The peak between 450 and 550 °C represents the oxidation of aromatic structures and organic compounds with great thermal stability [52]. Ino/CBW:CM ratio 2/0.7:0.3 was the digest that had the best performance in the degradation of the compounds compared with the Ino/CBW ratio 2:1 tests, where the peak intensity in the DTG at 550 °C was almost double that obtained in this test. Thus, this demonstrates that codigestion with CM in the ratio of 2/0.7:0.3 makes the CBW-to-AD process more effective for the treatment of effluent and the degradation of difficult compounds.

For the tests with FW, it was possible to verify that even with an inhibition in the production of biogas and biomethane, the results demonstrated a stabilization of the process. As with CM assays, peak values for compound degradation at a ratio of 2/0.7:0.3 were also more efficient.

This technique has characteristic bands of absorption of infrared radiation, which formed the basis for structural and qualitative analysis. The absorption bands have intensity directly proportional to the probability of the transition between fundamental and vibrational states [54]. The investigation of FTIR spectroscopy has been applied to the characterization of humic acids of different origins [55], as well as various organic residues from different treatments. The FTIR spectra found in digestion sludge show absorption bands linked to functional groups that depict the main chemical characteristics of the waste [56].

The FTIR analytical technique has been shown to be expeditious to verify which chemical compounds are present in samples, thus making it easier to verify the evolution of the composition of residues over the applied treatments. Authors have used this technique to evaluate the process they used [57–59].

The FTIR spectroscopy technique is a tool used to identify the main chemical groups that make up the sample. In this work, technology was used to verify the differences between the substrate and the inoculum used (Figure 5).

The vibrations in the band between 3750 and 3850 $cm^{-1}$ are related to the OH bonds [60]; the transmittance did not change in relation to the crude CBW [61]. As for the digested ones, the transmittance was reduced, demonstrating that the compounds were degraded. In all digested mixtures, it was possible to verify that the peak in the 3400 $cm^{-1}$ band corresponded to free or associated group bonds -OH and NH [62]. This band had undergone changes and was more evident, thus approaching Ino.

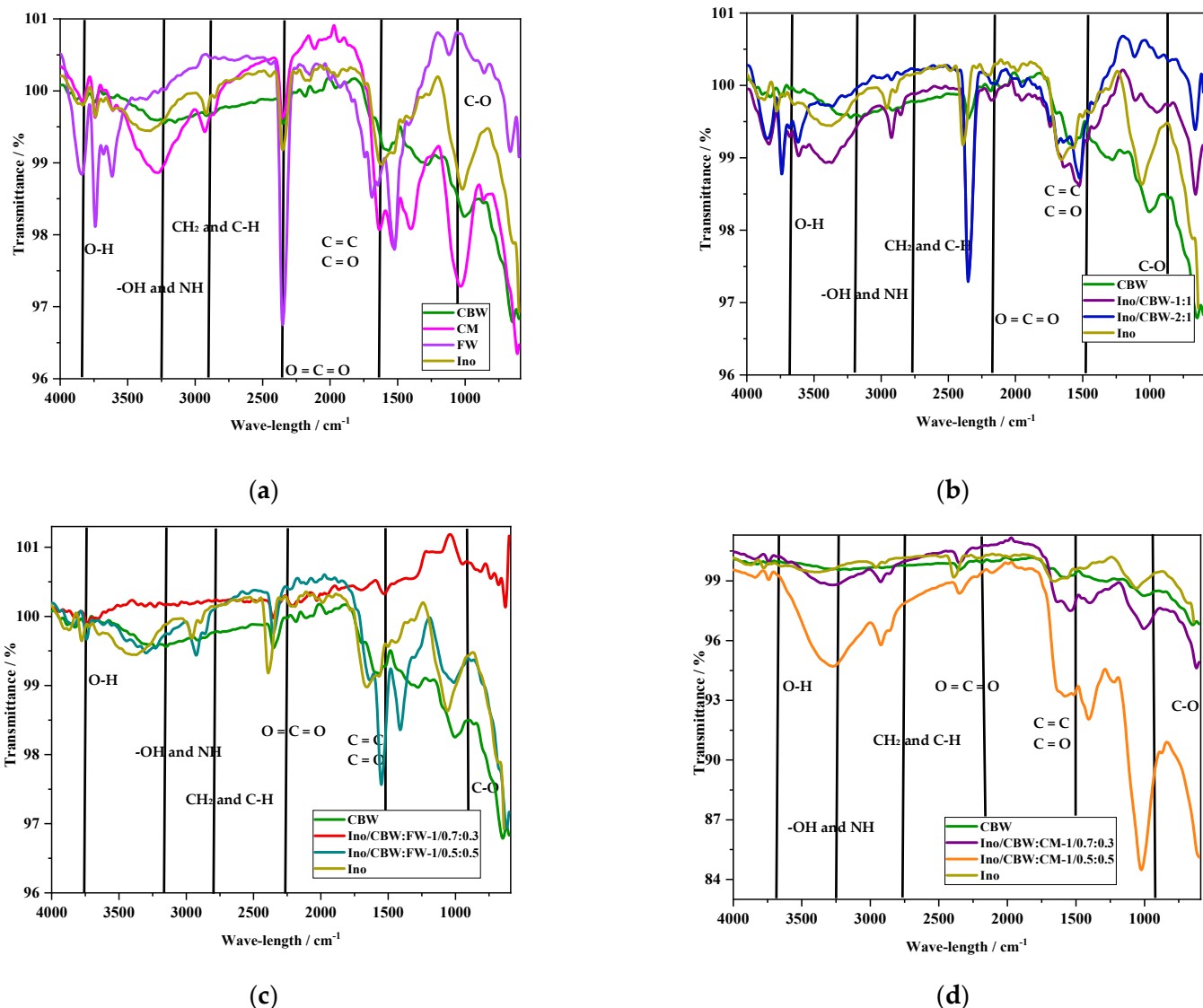

**Figure 5.** FTIR spectroscopy analysis. (**a**) Cork boiling wastewater (CBW), cow manure (CM), food waste (FW), and inoculum (Ino); (**b**) AD of different ratios of mixture of inoculum and cork boiling wastewater (Ino/CBW); (**c**) anaerobic digestion of different ratios of mixture of inoculum, cork boiling wastewater, and food waste (Ino/CBW:FW); (**d**) anaerobic digestion of different ratios of mixture of inoculum, cork boiling wastewater, and cow manure (Ino/CBW:CM).

The AD process generated a peak, with little impact, in the 2900 cm$^{-1}$ band. According to Domingos et al., the peak located in the 2900 cm$^{-1}$ band corresponds to the asymmetric CH$_2$ stretch [63] and also C–H bonds. The vibration region between 2300 and 2400 cm$^{-1}$ of stretching is very intense and, according to the literature, corresponds to the vibrations of CO$_2$ molecules [63,64], continuing with the transmittance at about 97%. The peak between 1630 and 1500 cm$^{-1}$, which is related to the aromatic rings C=C, C=O of amides, and acetones [59], was present in the process of DA for Ino/CBW for the ratios of 1:1 and 2:1; the average between repetitions showed that there were no changes in relation to the raw effluent. This peak in the other digestion assays appears intense, which, when compared with the digests of the codigestion, demonstrates the almost total degradation of the compounds for the Ino/CBW:FW assays and the Ino/CBW:CM assay for the ratio of 2/0.7:0.3.

The peak between 1630 and 1650 cm$^{-1}$, which is related to the C=C aromatic rings, [59] also corresponds to the conjugated, cyclic alkenes, α, β-unsaturated alkenes, and NH bonds

of the amines [64]. In the Ino/CBW:CM ratio of 2/0.5:0.5, there was no reduction in these compounds; for the other tests, these compounds were almost completely degraded.

In the band between 1200 and 1300 cm$^{-1}$ for OH deformations and elongations of CO bonds [59,63], this peak indicates the presence of phenols, and it was possible to verify that in the post AD effluent for Ino/CBW, this peak was no longer visible as it was in the gross CBW. In Ino/CBW:FW and Ino/CBW:CM codigerates, this peak was no longer visible. Peaks in the band between 1000 and 1100 cm$^{-1}$ were attributed to strains of CO bonds [65], which were also reduced for the Ino/CBW:FW tests and the Ino/CBW:CM test for the ratio of 2/0.7:0.3.

In a qualitative way, we can verify, when we compare the raw effluent with the two digested and studied inoculum, that the digestion process presents better stability to the digested mixture. The digested ones approached the stabilized inoculum, thus demonstrating that the AD process was able to degrade substances previously present in the initial effluent.

## 4. Conclusions

BMP tests have demonstrated that it may be possible to carry out the treatment of CBW effluent using the DA process. However, further studies are needed to conclude if the process is efficient by using different amounts of substrate and inoculum. In the tests, it was possible to verify that the final digestate had a pH close to neutrality.

For the Ino/CBW ratio of 1:1, methane production was $58.89 \pm 4.22$ mL.g.SV$_{added}$$^{-1}$, with a TS reduction of 13.97%. For the Ino/CBW ratio of 2:1, the production of methane was $99.18 \pm 2.03$ mL gSV$_{add}$$^{-1}$, showing a reduction of 6.81% of TS.

The tests show that the AD process of the effluent from cooking cork with the digestate of the WWTP in Castelo Branco-Portugal can be a viable alternative for the energetic use of this effluent and can assist in the stabilization of the effluent. Based on the FTIR tests that were carried out in a qualitative way, it is possible to verify that there was a reduction in the amount of phenols present in the effluent, thus granting greater stability to the effluent and facilitating later treatment. For more concrete conclusions regarding qualitative analysis and in order to use the effluent, it is necessary to analyze the digested mixture with more details in order to be able to verify the compounds present quantitatively.

The BMP codigestion tests demonstrated that the use of FW cosubstrates increased the production of biogas in relation to the CBW digestion tests. The use of FW as a cosubstrate for the energy recovery of CBW is not an advantage. With regard to the treatment of effluent, the use of FW is important because it helps in the degradation of complex compounds.

For the BMP tests of CBW codigestion with CM, biogas production values were found to be much higher. The stabilization of the CBW effluent was improved with the addition of CM for the codigestion process. The analyses of TGA, DTG, and FTIR showed that the use of Ino/CBW:CM at a ratio of 2/0.7:0.3 presented better treatment conditions. Methane production was lower, but the digestate was stabilized and is very close to the Ino used during the tests.

As evidenced by the possibility of being valued energetically, an optimization of the process to obtain greater amounts of biomethane production becomes feasible.

**Author Contributions:** Conceptualization, R.M.-P., M.J.H.-O., L.C.-C., P.S.D.d.B., and G.L.; methodology, R.M.-P., M.J.H.-O., L.C.-C., G.L., and P.S.D.d.B.; formal analysis, R.M.-P., L.C.-C., G.L., and P.S.D.d.B.; investigation R.M.-P., M.J.H.-O., L.C.-C., G.L., and P.S.D.d.B.; resources, P.S.D.d.B.; data curation, R.M.-P., L.C.-C., G.L., and P.S.D.d.B.; writing—original draft preparation, R.M.-P., L.C.-C., and P.S.D.d.B.; writing—review and editing, R.M.-P., L.C.-C., G.L., and P.S.D.d.B.; visualization, R.M.-P., L.C.-C., G.L., and P.S.D.d.B.; supervision, G.L. and P.S.D.d.B.; project administration, G.L. and P.S.D.d.B.; funding acquisition, G.L. and P.S.D.d.B. All authors have read and agreed to the published version of the manuscript.

**Funding:** This research received no external funding.

**Institutional Review Board Statement:** Not applicable.

**Informed Consent Statement:** Not applicable.

**Data Availability Statement:** Not applicable.

**Acknowledgments:** The authors are grateful for the financial support granted by the project 0049_INNOACE_4_E-Open and intelligent innovation at EUROACE, cofinanced by ERDF through the Interreg V-A Spain-Portugal program and the project ALT20-03-0145-FEDER-039485-SynDiesel, and for diesel fuels from dedicated waste and crop thermal gasification, cofinanced by ERDF through the Regional Operational Program of the Alentejo. G. Lourinho also acknowledges FCT (Fundação para a Ciência e Tecnologia) for financial support under grant SFRH/BDE/111878/2015.

**Conflicts of Interest:** The authors declare no conflict of interest.

**Novelty Statement:** CBW is an effluent that is difficult to treat due to its characteristics (low biodegradability and acidic pH). BMP tests were carried out using different proportions of only the CBW and an inoculum, with the aim of verifying whether it is possible to carry out a biological treatment on the effluent.

## Abbreviations

| | |
|---|---|
| AD | Anaerobic Digestion |
| BMP | Biochemical Methane Potential |
| BOD | Biochemical Oxygen Demand |
| C | Carbon |
| CBW | Cork Boiling Wastewater |
| CH4 | Methane |
| CM | Cow Manure |
| CO | Carbon monoxide |
| CO2 | Carbon Dioxide |
| COD | Chemical Oxygen Demand |
| DNA | Deoxyribonucleic Acid |
| DTG | Differential Thermal Analyzes |
| FTIR | Fourier transformed infrared |
| FW | Food Waste |
| H | Hydrogen |
| H2S | Hydrogen Sulfide |
| Ino/CBW | Inoculum/Cork Boiling Wastewater |
| N | Nitrogen |
| O2 | Oxygen |
| POA | Advanced Oxidation Processes |
| S | Sulfur |
| TGA | Thermal Gravimetric Analysis |
| TS | Total Solids |
| VFA | Volatile Fatty Acids |
| VS | Volatile Solids |
| WWTP | Wastewater Treatment Plant |

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
