# Peer review of "Biochemical Methane Potential of Cork Boiling Wastewater at Different Inoculum to Substrate Ratios"

_applsci, doi:10.3390/app11073064_

Round 1

Reviewer 1 Report

The work entitled “Biochemical methane potential of cork boiling wastewater at different inoculum to substrate ratios” by the Panizio has been reviewed. The work described that biomethane potential of CBWs for Ino/CBW ratios 1:1 and 2:1 is very low when compared to other organic substrates. e Ino/CBW:CM ratio 2/0.7:0.3 test shows better biomethane yields, being in the expected values and demonstrating that it is possible to perform CBW the AD using a cosubstrate to increase biogas production and biomethane, as well as improving the quality of the final digestate.

The abstract need to clearly state the objective and background of the study with M&M and findings of the work.  In the present abstract is difficult to understand the work for leaders.

The introduction section should be converted as paragraph rather than the points. Besides, the introduction section well written.

M&M a subtitle deals with the materials used in the study needs to be provided.

The information about the inoculum of AD is too general, which need the more specific information.

Figure. Legends should be explained more details by simple definition with explanation of the abbreviations.

Some figure axis value are not clear in visibility, authors are advised to improve the figure quality.

Table 1, Table 2 data need to present with mean ±SE, with post hoc test results.

Fig.1, 2,3 need to provide the mean ±SE based graph.

Fig 3 legend is completely confusing need to revised.

Fig.4 FTIR results need to provided details legend for each graph with highlighting important shifting molecules.

Author Response

The work entitled “Biochemical methane potential of cork boiling wastewater at different inoculum to substrate ratios” by the Panizio has been reviewed. The work described that biomethane potential of CBWs for Ino/CBW ratios 1:1 and 2:1 is very low when compared to other organic substrates. e Ino/CBW:CM ratio 2/0.7:0.3 test shows better biomethane yields, being in the expected values and demonstrating that it is possible to perform CBW the AD using a cosubstrate to increase biogas production and biomethane, as well as improving the quality of the final digestate.

The abstract need to clearly state the objective and background of the study with M&M and findings of the work.  In the present abstract is difficult to understand the work for leaders.

Answer: The abstract has been changed and reduced, thus making it more simplified and easy to understand

The introduction section should be converted as paragraph rather than the points. Besides, the introduction section well written.

Answer: were converted to paragraph.

M&M a subtitle deals with the materials used in the study needs to be provided.

Answer: The subtitle "2.1 Substrate, inoculum and co-substrates" has been removed.

The information about the inoculum of AD is too general, which need the more specific information.

Answer: Line 194-198

“The inoculum (Ino) used for BMP tests obtained from an anaerobic digester in a wastewater treatment plant in a city in central Portugal. The treatment unit has an anaerobic digester in operation and Ino was collected from the end of the AD process, thus containing large amounts of bacteria, thus facilitating the start of operation of the studied digesters. Ino is an established digest and ready to be used in other AD tests or to be treated at the wastewater treatment plant.”

Figure. Legends should be explained more details by simple definition with explanation of the abbreviations.

Have been modified

Some figure axis value are not clear in visibility, authors are advised to improve the figure quality.

Have been modified

Table 1, Table 2 data need to present with mean ±SE, with post hoc test results.

Answer: Standard deviations have been added

Fig.1, 2,3 need to provide the mean ±SE based graph.

Have been modified

Fig 3 legend is completely confusing need to revised.

Have been modified

Fig.4 FTIR results need to provided details legend for each graph with highlighting important shifting molecules.

The peaks have been identified in the graph.

Reviewer 2 Report

The authors have presented an interesting topic that is worth publishing. Although, the writing needs significant improvement. I recommend detailed corrections by the author. Other manuscripts in the research field can be referred as examples to understand formatting and writing style.

Author Response

Dear Reviewer. Thank you very much for your constructive review.
We have reviewed our manuscript especially in the introductory part, referring to other similar research manuscripts. Thanks a lot

Reviewer 3 Report

Biochemical methane potential test is a widely used method in the science and industry practice, as well. Determination of the biogas potential of different materials can provide useful information for the science, and practice, as well. The ratio of the inoculum to substrate (wastewater or sludge, for instance) is one of the key issues contribute to achieve good repeatability in the BMP analysis.

Biodegradability af cork boiling wastewater has not been investigated in details, yet. As author mentioned the cork boiling wastewater has a low biodegradability because it contains several components with inhibition effect on aerobic and anaerobic degradation. Therefore the manuscript applsci-1159971 contains interesting information for the readers.

The manuscript is generally well written with a logic structure. Introduction summarized well the solvende problems, the theoretical background of the research and the research motivations and the main goals of the research. Materials and methods are described clearly. Manuscript contains interesting and valuable results, which are discussed together with the results and experiences oof relevant references.

Comments, suggestions:

Table 1 and Table 2 do not contain the standard deviations.

There can be seen the difference of cumulative biogas production in Figure 1 and 2, but, in my opinion, the presentation of specific biogas production values (biogas volume/VS or TS, for instance) would be more informative and made the results more comparable.

The C/N ratio has effect on biogas production yield. This parameter has great importance for co-digestion, as well.  I suggest the authors to discuss this aspect, and calculate this parameter for the samples (elemental analysis results are given in the manuscript) and discuss this aspect related to their experimentally determined biogas production values.

Author Response

Biochemical methane potential test is a widely used method in the science and industry practice, as well. Determination of the biogas potential of different materials can provide useful information for the science, and practice, as well. The ratio of the inoculum to substrate (wastewater or sludge, for instance) is one of the key issues contribute to achieve good repeatability in the BMP analysis.

Biodegradability af cork boiling wastewater has not been investigated in details, yet. As author mentioned the cork boiling wastewater has a low biodegradability because it contains several components with inhibition effect on aerobic and anaerobic degradation. Therefore the manuscript applsci-1159971 contains interesting information for the readers.

The manuscript is generally well written with a logic structure. Introduction summarized well the solvende problems, the theoretical background of the research and the research motivations and the main goals of the research. Materials and methods are described clearly. Manuscript contains interesting and valuable results, which are discussed together with the results and experiences oof relevant references.

Comments, suggestions:

Table 1 and Table 2 do not contain the standard deviations.

Answer: Standard deviations were added for the data shown in tables 1 and 2.

There can be seen the difference of cumulative biogas production in Figure 1 and 2, but, in my opinion, the presentation of specific biogas production values (biogas volume/VS or TS, for instance) would be more informative and made the results more comparable.

Answer: Thanks for the excellent suggestion, the graphics presented with the degradation of the VS would really be interesting. However, the bottles were sealed and it was not possible to carry out the collection of the effluent during the tests to carry out the monitoring of the biogas production in relation to the degradation of the VS.

The C/N ratio has effect on biogas production yield. This parameter has great importance for co-digestion, as well.  I suggest the authors to discuss this aspect, and calculate this parameter for the samples (elemental analysis results are given in the manuscript) and discuss this aspect related to their experimentally determined biogas production values.

Answer: 284 - 289

“The C/N ratio reflects the availability of nutrients in CBW, Ino, FW and CM for the growth of bacteria. The low C/N ratio shows that the microorganism does not have adequate amounts of nutrients, especially carbon, for them to develop. Wang et al. (2014) reported that the low C/N ratio can result in the formation of large amounts of ammonia, which is toxic to the bacterial population, since nitrite accumulates during the processes of biological denitrification, thus causing the inhibition of the process [40].”

Round 2

Reviewer 1 Report

Accept

Author Response

Dear reviewer:
Thank you very much for your constructive review and acceptance of our work.
Thanks a lot
Kind regards

Reviewer 2 Report

Dear Author,

Please go through my comments in the word file. I shall be glad if you could write to my comments in detail, as well as make necessary updates to the manuscript.

Author Response

The authors present an interesting work on biochemical methane potential test on cork boiling wastewater. The major traits covered in this article, include characterization of wastewater, food waste and Inoculum, and their digestate. The author also discusses thermogravimetric and spectroscopic analysis results. Overall, the authors have done a great job in this work. But there are many presentation flaws. I recommend the author to rectify the below mentioned ones, to strengthen the manuscript.

  1. Abbreviations must be listed. Without a list of abbreviations, it is difficult for the reader to fetch for the abbreviation expansion every time.

Answer: A list of abbreviations has been added.

  1. There are too many abbreviations in the manuscript. It is very difficult to follow and understand the information due to high number of abbreviations. The author should instead use the full text. The author should be considerate in utilizing abbreviations for texts.

Answer: A list of abbreviations has been added to make it easier to find abbreviations and understand them throughout the manuscript

  1. Line 23: The representation of “2/7:0.3” as ratio, is confusing. Consider using an alternative for all similar texts.

Answer: Prior to said "2 / 0.7: 0.3" it is always indicating which material it is referring to. The use of generic names for each experiment can confuse the reader. I try the front of each numerical reference which is the relative mixture, it tends to facilitate the perception.

  1. Please check if all the units are in SI system

Answer: Changed as requested.

  1. Line 26, 307: gVSadded-1 is not clear. What does “VSadded-1” mean?

Answer: Changed as requested

  1. Line 38: Keywords should have consistent formatting (all small letters)

Answer: Changed as requested

  1. Lines 42 – 44: Are these part of abstract? Please justify.

Answer: No, it refers to the declaration of novelty that we have always presented.

  1. Line 61: “about 90%” the sentence is incomplete. 90% of?

Answer:  corrections were made. Line 54-55

“Phenolic compounds represent about 90% of contaminants, present in the CBW [5].”

  1. Line 63: “The process”. what does the process refer to?
  2. The author should consider combining shorter paragraphs. One larger paragraph delivering a common knowledge instead of many smaller para with related information.
  3. Line 72: Details on cork boiling wastewater can be included. Information including annual production, current status is lacking.

Answer: Changed as requested. Line 59 to 65.

“It is estimated that in Portugal around 34% of the total area is currently explored, which corresponds to an average annual production of 150 thousand tons. The cork transformation process involves the use of large amounts of water and, consequently, a high production of waste water. It is estimated that 400 L of water are consumed per ton of cork processed in the raw material purification stage, known as cooking, which has as main objective the purification of the raw material, which is immersed in hot water (temperature 95 to 105ºC) for one hour. These 150 thousand tons of cork, originates about 6.0 x 107 L of CBW”

  1. Line 72: “CBO5/COD” clarify “5”?

Answer: When referring to BOD5, it states that BOD was measured at the end of 5 days.

  1. Line 84: “previous” – A different word has to be used instead

Answer: Changed as requested

  1. Line 158: “diffe.rent” – Make necessary changes

Answer: Changed as requested

  1. Line 199: specify country

Answer: Changed as requested

  1. Line 209: quantification on Food waste, cow manure and other substrates should be included.

Answer: The values were quantified in terms of proportions of added VS, as described in lines 183 to 186.

  1. Lines 218 – 228: The methods can be explained and cited

Answer:

Dear Reviewer,

You are right in the observation however we do not explain in detail as they were based on the Standard methodology which is a methodology used worldwide and would make it repetitive to explain again since a reference has already been made.

  1. Line 240: “1.000 ml” Inconsistent decimal places

Answer: You are right, the correct value is 1000 mL. It has already been corrected in the manuscript

  1. The literature review covers many different work performed. But detailed information on cork boiling water (origination, treatment methods are lacking). The author should also consider the relevance of literature information (page 3 and 4). Data that lies outside the scope should be neglected.

Answer: The works that are studied using the cork effluent for AD are present in the article. When it comes to the study of cork, it is basically limited to the Iberian peninsula and Morocco, which makes it difficult to find jobs in this area. With regard to AD, it becomes an innovative and very specific work because in the literature, it is only possible to find an author who studied this effluent for digestion

  1. Section 2.1: Details on anerobic digestion methodology are lacking. The author should provide detailed information on quantity of CNW, FW and CM, operating conditions and other parameters. Providing more data about the experiments will help the reader to better understand and adapt their protocol in future studies.

Answer: The details of the tests are specified between lines 240 to 258, in section 2.2.

  1. The text in tables should be left aligned

Answer: Changed as requested

  1. Line 292 – “Concerning” – The author should consider rewording 23.Subdivide the charts in figures into a,b,c. refer to them in text as fig 1 (A)

Answer: Changed as requested

  1. Table 2. “%VS” – Does this mean wt%?

Answer: Thanks for the correction. The percentage means by weight

  1. The results and discussion are very long. The author should consider moving trivial information to supplementary.

Answer: Changed as requested

Round 3

Reviewer 2 Report

Dear Author - Thank you for updating the manuscript.

This manuscript is a resubmission of an earlier submission. The following is a list of the peer review reports and author responses from that submission.

Round 1

Reviewer 1 Report

The authors investigated the Biochemical methane potential of cork boiling wastewater at different inoculum to substrate ratios. In my opinion, the manuscript can be accepted after major revision.

Some critical comments for authors.

  1. In the abstract, some abbreviations are not explained. For example, WWTP, ISR, STP, and AD. Space between the words should be corrected.
  2. The novelty statement should be written clearly.
  3. Line no. 51. Should be written “However, its composition depends on many variables such as climatic conditions, soil conditions, origin, dimensions, and age of trees”.
  4. Line no. 53. The extracts present in cork are non‐structural organic components of the cellular appearance. What author does mean?
  5. “During the process of cooking cork boards, a wide range of organic compounds is transferred to boiling water, thus resulting in a generation of wastewater with a high organic load and little biodegradability”. How authors made this statement? Why biodegradable materials having non-biodegradable things and even in the water extract.
  6. The cork boiling wastewater, despite having a low biodegradability due to the presence of polyphenols and tannins, currently no specific treatments are applied to these waters before discharge into the wastewater treatment plants (WWTP). What the authors mean is solubility or biodegradability? If it is biodegradability author should provide additional information to prove their statement.
  7. Please explain about inoculum. What are things contained in it?
  8. Line no. 252. What author does mean co-substrate? Not mentioned in the method section.
  9. Line no 255. COD mentioned as 7060.0 mg/mL, but in Table 1. COD mentioned as 7.060 mg/L.???
  10. How the author can conclude the cellulose and lignin degradation using thermogravimetric analysis and FTIR. How does the author characterize those molecules in it? What is the need for thermogravimetric analysis?
  11. Why the section “Thermogravimetric and spectroscopic analysis” repeated? Is there any difference between sections 3.2 and 3.5?
  12. Authors are advised to double-check the values and typographical errors throughout the manuscript.

Author Response

Some critical comments for authors.

  1. In the abstract, some abbreviations are not explained. For example, WWTP, ISR, STP, and AD. Space between the words should be corrected.

Corrections were made as requested and are highlighted in the text.

  1. The novelty statement should be written clearly.

“CBW is an effluent that is difficult to treat due to its characteristics (low biodegradability and acidic pH). BMP tests were carried out with the aim of verifying whether it is possible to carry out a biological treatment on these effluents. For the tests, different proportions were performed, with only the CBW and an inoculum.”

  1. Line no. 51. Should be written “However, its composition depends on many variables such as climatic conditions, soil conditions, origin, dimensions, and age of trees”.

Changed.

  1. Line no. 53. The extracts present in cork are non‐structural organic components of the cellular appearance. What author does mean?

It is understood by cell wall and not cellular aspect. Was changed.

  1. “During the process of cooking cork boards, a wide range of organic compounds is transferred to boiling water, thus resulting in a generation of wastewater with a high organic load and little biodegradability”. How authors made this statement? Why biodegradable materials having non-biodegradable things and even in the water extract.

This statement was taken from an article that is properly referenced. The effluent from the cork cooking process has many biodegradable compounds, however all authors who have studied this effluent cite that it has low biodegradability from the phenolic compounds present in the effluent. Presenting a low CBO5/COD ratio.

  1. The cork boiling wastewater, despite having a low biodegradability due to the presence of polyphenols and tannins, currently no specific treatments are applied to these waters before discharge into the wastewater treatment plants (WWTP). What the authors mean is solubility or biodegradability? If it is biodegradability author should provide additional information to prove their statement.

The statement is correct, the low biodegradability mentioned is related to the CBO5/COD ratio that other authors have studied and found to be between 0.3 and 0.45.

  1. Please explain about inoculum. What are things contained in it?

“The inoculum (INO) used for BMP tests came from an anaerobic digester at a wastewater treatment plant in a city in central Portugal. The treatment unit has an anaerobic digester in operation and the digested was supplied due to having large amounts of bacteria, thus facilitating the start of operation of the studied digesters.”

  1. Line no. 252. What author does mean co-substrate? Not mentioned in the method section.

In this study, there was no use of co-substrate. Thank you for your careful reading. It has been removed from the text.

  1. Line no 255. COD mentioned as 7060.0 mg/mL, but in Table 1. COD mentioned as 7.060 mg/L.???

The correct value is 7060.00 mg / L. The same has been corrected in the table.

  1. How the author can conclude the cellulose and lignin degradation using thermogravimetric analysis and FTIR. How does the author characterize those molecules in it? What is the need for thermogravimetric analysis?

“The TGA and DTG analyzes make it possible to know for each sample the weight loss that occurs in a given temperature range during its combustion. The higher the temperature at which weight loss occurs, the more resistant and structurally organized is the organic fraction that is burning. Comparing different groups with different degrees of stability, according to the temperature changes in which the main weight loss occurs during combustion should be indicative of the characteristics of the organic fraction. On the other hand, in comparison with the initial sample, whenever a weight loss disappears, in the TGA and DTG profiles, a temperature at which it was presented in the crude analyzes, may indicate that the corresponding organic fraction that was being burned was mineralized due to the stabilization process. This weight loss may consider the AD process to be efficient [34] [35] [36].”

  1. Why the section “Thermogravimetric and spectroscopic analysis” repeated? Is there any difference between sections 3.2 and 3.5?

The two sections were put together. They were separated just by talking about the digestate. But section 3.2 already contains information about the substrate, inoculum and the two digested.

  1. Authors are advised to double-check the values and typographical errors throughout the manuscript.

Corrections were made as requested.

Reviewer 2 Report

The authors investigated biochemical methane potential of cork boiling wastewater at different inoculum to substrate ratios. Some results seem to be interesting, but they determined BMP at only two ISR conditions. I think the novelty and amount of data are not sufficient to be published in Applied Science. I have follow specific comments.

1. Current introduction is too general. I strongly recommend that re-organize the introduction part emphasizing the research novelty.

2. Figure 1

Please improve the resolution.

3. Table 1

Probably, the data for CBW is wrong. For example, the COD concentration was written by 7060.0 mg/L in the manuscript, but it was reported by 7.060 mg/L in Table 1.

4. Figure 4

Please add standard deviation.

5. 3.2 Thermogravimetric and spectroscopic analysis, 3.5 Thermogravimetric and spectroscopic analysis

Both sub-chapters are less related to the research objective, assessing the BMP of cork boiling wastewater.

6. Line 424

If phenol is important parameter, the authors should evaluate phenol degradation, quantitatively.

Author Response

The authors investigated biochemical methane potential of cork boiling wastewater at different inoculum to substrate ratios. Some results seem to be interesting, but they determined BMP at only two ISR conditions. I think the novelty and amount of data are not sufficient to be published in Applied Science. I have follow specific comments.

  1. Current introduction is too general. I strongly recommend that re-organize the introduction part emphasizing the research novelty.

The introduction has been changed. Note that the following sentence was added at the end of the introduction:

“The cork effluent is very problematic due to the acidic character and constituents of low degradability, thus making it difficult to treat. Currently, few studies of cork boiling wastewater treatment by AD have been carried out and the current information is very limited. The authors are only aware of a previous study that evaluated the AD of this effluent in order to enhance the reduction of organic pollutants [21]. Based on the considerations above, the present work aims to investigate the effect of two different ratios of Ino/S (1: 1 and 2: 1) on the yield of accumulated methane production and on the degradation of VS.”

  1. Figure 1

Please improve the resolution.

Resolution has been fixed.

  1. Table 1

Probably, the data for CBW is wrong. For example, the COD concentration was written by 7060.0 mg/L in the manuscript, but it was reported by 7.060 mg/L in Table 1.

The correct value is 7060.00 mg / L. The same has been corrected in the table. 

  1. Figure 4

Please add standard deviation.

The standard deviation was cited in the text. 

  1. 3.2 Thermogravimetric and spectroscopic analysis, 3.5 Thermogravimetric and spectroscopic analysis

Both sub-chapters are less related to the research objective, assessing the BMP of cork boiling wastewater.

The chapters have been joined. They are presented only as a way to verify that there have been changes and have degraded some substance. Due to restricted conditions in terms of the laboratory, these analyzes were used as a way of verifying whether the digestion process made any changes in the effluent conditions. The studies were based on other studies that they used to be able to verify if the effluent has stabilized. The statement was substantiated with the following paragraph.

“The TGA and DTG analyzes make it possible to know for each sample the weight loss that occurs in a given temperature range during its combustion. The higher the temperature at which weight loss occurs, the more resistant and structurally organized is the organic fraction that is burning. Comparing different groups with different degrees of stability, according to the temperature changes in which the main weight loss occurs during combustion should be indicative of the characteristics of the organic fraction. On the other hand, in comparison with the initial sample, whenever a weight loss disappears, in the TGA and DTG profiles, a temperature at which it was presented in the crude analyzes, may indicate that the corresponding organic fraction that was being burned was mineralized due to the stabilization process. This weight loss may consider the AD process to be efficient [34] [35] [36].”

  1. Line 424

If phenol is important parameter, the authors should evaluate phenol degradation, quantitatively.

Phenols and polyphenols are important in these effluents, the authors acknowledge. However, the analyzes performed were qualitative through FTIR, like many other works. Due to equipment limitations, we carry out the quantitative based on other works.

Reviewer 3 Report

Biochemical methane potential of cork boiling wastewater was evaluated at the inoculum to substrate ratios of 1:1 and 2:1. First of all, the experimental set up should have contained more ISR combinations, only 2 ratios do not provide sufficient info. Then, too many aspects are missing (statistical analysis, chemical analysis (i.e. phenols)) There are too many vague intentions (i.e. FTIR analysis of the slurries) and results that even the authors did not discuss. Unfortunately, the manuscript do not meet the requirements for publishing.

Introduction needs major revision and to be re-written.

L21: Please omit “adequate”, and add on the basis of VS after the ISRs.

L24-25: No need for such well-known info “calculated by dividing the final methane volumes by the weight (in VS) of substrate added to the reactors” in the abstract.

L53: First 4 paragraphs can be combined. There are too many separated paragraphs that are related to each other.

L76-78: Please revise this sentence, so many commas.

L83-145: Half of the Introduction belongs to other several treatment methods that are not related to the study. At the end this is not a review paper. I suggest the authors to summarise this section in few lines, and to focus on anaerobic digestion.

L168-172: This is too general info, should be omitted. Please mention the importance of ISR. The introduction lacks some related important references. Here are few recent suggestions to be included in the paper:

  • Li et al., 2018 Influence of feed/inoculum ratios and waste cooking oil content on the mesophilic anaerobic digestion of food waste
  • Cordoba et al., 2018 The effect of substrate/inoculum ratio on the kinetics of methane production in swine wastewater anaerobic digestion
  • Akyol 2020 In search of the optimal inoculum to substrate ratio during anaerobic co-digestion of spent coffee grounds and cow manure
  • Hobbs et al., 2018 Enhancing anaerobic digestion of food waste through biochemical methane potential assays at different substrate: inoculum ratios
  • Cremonez et al. 2019 Influence of inoculum to substrate ratio on the anaerobic digestion of a cassava starch polymer

L173-176: The last paragraph should be enhanced; the objective and the novelty should be better highlighted.

L178-176: This part should be under “Section 2.1 Substrate and inoculum”

L185-186: Please use “obtain” instead of “come” for the materials.

L197-198: No need to repeat

L205-210: No need to explain a very well known procedure

L212: Please indicate the full name for the first time.

L225: Please use one abbreviation for the ISR. Is this VS basis?

L225: Which inert gas?

Figure 1 should be omitted. Too general and nothing specific to the study.

L252: Other co-substrates are referred to other studies? Not clear. Please first present your results/data, then discuss other studies.

L250-254: In the Introduction, many compounds are mentioned for CBW (i.e. phenolic compounds, tannins), but none of them were analysed.

L269-270: This is just speculation…

L275-303: What is the important result here? I could not get the point…

L321: What is the reason for FTIR analysis of slurries?

L354-357: Are these differences significant? No statistical analysis…

L359-363: So, what are the results telling us? No discussion…

L366: Why another same-titled sub-section?

L424: When you only say that phenols are present in CBW, which is a fact, and don’t analyse phenols, how can you conclude this in your own conclusion?

Author Response

Biochemical methane potential of cork boiling wastewater was evaluated at the inoculum to substrate ratios of 1:1 and 2:1. First of all, the experimental set up should have contained more ISR combinations, only 2 ratios do not provide sufficient info. Then, too many aspects are missing (statistical analysis, chemical analysis (i.e. phenols)) There are too many vague intentions (i.e. FTIR analysis of the slurries) and results that even the authors did not discuss. Unfortunately, the manuscript do not meet the requirements for publishing.

Introduction needs major revision and to be re-written.

L21: Please omit “adequate”, and add on the basis of VS after the ISRs.

The text was corrected according to the suggestion.

L24-25: No need for such well-known info “calculated by dividing the final methane volumes by the weight (in VS) of substrate added to the reactors” in the abstract.

The text was corrected according to the suggestion.

L53: First 4 paragraphs can be combined. There are too many separated paragraphs that are related to each other.

The text was corrected according to the suggestion.

L76-78: Please revise this sentence, so many commas.

The text was corrected according to the suggestion.

L83-145: Half of the Introduction belongs to other several treatment methods that are not related to the study. At the end this is not a review paper. I suggest the authors to summarise this section in few lines, and to focus on anaerobic digestion.

The reviewer is right, the text has been changed and summarized

L168-172: This is too general info, should be omitted. Please mention the importance of ISR. The introduction lacks some related important references. Here are few recent suggestions to be included in the paper:

  • Li et al., 2018 Influence of feed/inoculum ratios and waste cooking oil content on the mesophilic anaerobic digestion of food waste
  • Cordoba et al., 2018 The effect of substrate/inoculum ratio on the kinetics of methane production in swine wastewater anaerobic digestion
  • Akyol 2020 In search of the optimal inoculum to substrate ratio during anaerobic co-digestion of spent coffee grounds and cow manure
  • Hobbs et al., 2018 Enhancing anaerobic digestion of food waste through biochemical methane potential assays at different substrate: inoculum ratios
  • Cremonez et al. 2019 Influence of inoculum to substrate ratio on the anaerobic digestion of a cassava starch polymer

The text was revised according to the suggestion, and references were added.

L173-176: The last paragraph should be enhanced; the objective and the novelty should be better highlighted.

The text was corrected according to the suggestion.

L178-176: This part should be under “Section 2.1 Substrate and inoculum”

Section 2 was modified as suggested.

L185-186: Please use “obtain” instead of “come” for the materials.

Phrase has been modified

L197-198: No need to repeat

It has been removed from the text

L205-210: No need to explain a very well known procedure

Only the Standard Methods used were created

L212: Please indicate the full name for the first time.

The name was indicated

L225: Please use one abbreviation for the ISR. Is this VS basis?

Yes. It was added in the text. Line 237.

L225: Which inert gas?

The inert gas used was Argon, Line 238

Figure 1 should be omitted. Too general and nothing specific to the study.

Figure 1 was removed as requested

L252: Other co-substrates are referred to other studies? Not clear. Please first present your results/data, then discuss other studies.

In this study, there was no use of co-substrate. Thank you for your careful reading. It has been removed from the text.

L250-254: In the Introduction, many compounds are mentioned for CBW (i.e. phenolic compounds, tannins), but none of them were analysed.

The analyzes were not performed because the objective of the work was to determine if the effluent has the potential to produce methane. We know that these analyzes are important but for the present work only qualitative analyzes were carried out for polyphenols and phenols, through FTIR.

L269-270: This is just speculation…

The reviewer is right, it is a speculation, but at the level of comparison the effluents of the cork are very variable because they are dependent on the region from which the cork is extracted and thus gives it distinct characteristics. Thus, comparing an effluent from a different region and effluents from the same region, becomes more reliable because despite the different batches, the characteristics of the board treatment process are similar. Changed to (line 266 to 271):

“Ponce-Robles et al. used in their tests the effluents from the association plant of the cork industries of San Vicente de Alcântara. Despite being from different lots, the pH (5.0) and conductivity (1684 µS/cm) values ​​are relatively close. The authors characterized the amount of polyphenols present in the effluent, being in the order of 455 mg/L. As they are effluents from the same region and produced by the same companies, it is possible that despite being from different lots, the conditions are as similar as possible when compared to other effluents.”

L275-303: What is the important result here? I could not get the point…

The text has been changed. Line 275 to 327.

L321: What is the reason for FTIR analysis of slurries?

Line 331 to 342:

This technique has characteristic bands of absorption of infrared radiation, which formed the basis for structural and qualitative analysis. The absorption bands have intensity directly proportional to the probability of the transition between the fundamental and vibrational states [43]. The investigation of FTIR spectroscopy has been applied to the characterization of humic acids of different origins [44], as well as various organic residues from different treatments. The FTIR spectra found in digestion sludge show absorption bands linked to functional groups that depict the main chemical characteristics of the waste [45].

The FTIR analytical technique has been shown to be expeditious to verify which chemical compounds are present in samples, thus making it easier to verify the evolution of the composition of residues over the applied treatments. Authors used this technique to evaluate the process they used [46] [47] [48].

L354-357: Are these differences significant? No statistical analysis…

A figure has been added for better interpretation.

L359-363: So, what are the results telling us? No discussion…

Line 387 to 390.

“In a qualitative way, we can verify if we compare the raw effluent with the two digested and the studied inoculum that the digestion process presented a better stability to the digested. The digested ones approached the stabilized inoculum, thus demonstrating that the AD process was able to degrade substances previously present in the initial effluent”

L366: Why another same-titled sub-section?

The sections were joined.

L424: When you only say that phenols are present in CBW, which is a fact, and don’t analyse phenols, how can you conclude this in your own conclusion?

Line to 425 to 432.

“The tests showed that the AD process of the effluent from cooking cork with the digestate of the WWTP in Castelo Branco, can be a viable alternative for the energetic use of this effluent and assist in the stabilization of the effluent. Based on the FTIR tests that were carried out, in a qualitative way it is possible to verify that there was a reduction in the amount of phenols present in the effluent, thus granting a greater stability to the effluent and may come to facilitate later treatment. For more concrete conclusions regarding qualitative analysis and in order to use the effluent, it is necessary to analyze the digested with more details in order to be able to verify quantitatively the compounds present.”

Round 2

Reviewer 1 Report

accept

Reviewer 3 Report

The authors addressed most majority of the results. Moderate English revision is required.